# Proteome Analysis Reveals Syndecan 1 Regulates Porcine Sapelovirus Replication

**DOI:** 10.3390/ijms21124386

**Published:** 2020-06-19

**Authors:** Tingting Zhao, Li Cui, Xiangqian Yu, Zhonghai Zhang, Qi Chen, Xiuguo Hua

**Affiliations:** 1Shanghai Key Laboratory of Veterinary Biotechnology, School of Agriculture and Biology, Shanghai Jiao Tong University, Shanghai 200240, China; ztt0407@sjtu.edu.cn (T.Z.); lcui@sjtu.edu.cn (L.C.); 2Shanghai Pudong New Area Center for Animal Disease Control and Prevention, Shanghai 200136, China; yuxiangqian134@126.com (X.Y.); zhonghaiz@126.com (Z.Z.); 3Shanghai Animal Disease Control Center, Shanghai 201103, China; shdwyk@163.com

**Keywords:** quantitative proteomics, tandem mass tag, sapelovirus, replication

## Abstract

Porcine sapelovirus A (PSV) is a single stranded, positive-sense, non-enveloped RNA virus that causes enteritis, pneumonia, polioencephalomyelitis, and reproductive disorders in pigs. Research on PSV infection and interaction with host cells is unclear. In this study, we applied tandem mass tag proteomics analysis to investigate the differentially expressed proteins (DEPs) in PSV-infected pig kidney (PK)-15 cells and explored the interactions between PSV and host cells. Here we mapped 181 DEPs, including 59 up-regulated and 122 down-regulated DEPs. Among them, osteopontin (SPP1), induced protein with tetratricopeptide repeats 5 (IFIT5), ISG15 ubiquitin-like modifier (ISG15), vinculin (VCL), and syndecan-1 (SDC1) were verified significantly changed using RT-qPCR. Additionally, overexpression of SDC1 promoted PSV viral protein (VP)1 synthesis and virus titer, and silencing of SDC1 revealed the opposite results. Our findings show that SDC1 is a novel host protein and plays crucial roles in regulating PSV replication.

## 1. Introduction

Porcine sapelovirus A (PSV), a member of the genus sapelovirus in the family *Picornaviridae*, is a single-stranded, positive-sense, non-enveloped RNA virus with a full-length genome of 7.5–8.3 nucleotide (nt) that contains a 3′ poly (A) tail with a variable length from 65 to 100 nt. Four structural proteins (viral protein (VP)1, VP2, VP3, and VP4) constitute the virus capsid which mediates the initiation of infection by binding to a receptor on the host membrane, and seven non-structural proteins (2B–2D, 3A–3D) participate in virus replication [1]. Diseases caused by PSV are characterized mainly by acute diarrhea, polioencephalomyelitis, reproductive disorders, and even fatality in pigs [2,3,4]. The virus can be amplified and secreted in pig kidney (PK)-15 cells, IBRS-2 and LLC-PK [4,5] and human hepatocarcinoma cells PLC/PRF/5 and HepG2/C3a [6]. A previous study reported that PSV can use α-2,3-linked sialic acid on GD1a ganglioside as a receptor entry into LLC-PK1 cells [5] and PSV enters its host cells via caveolae-dependent endocytosis independent of clathrin-coated vesicle [7]. However, research on PSV is mainly focused on epidemiological investigations and the mechanism of PSV replication and pathogenesis has not been fully understood.

Gaining more information about PSV biology to find potential targets in its replicative cycle might provide help in the development of appropriate strategies to control PSV infection. Therefore, identification of host proteins, which viruses use during their replicative cycle, is essential. Recently, many researchers have successfully utilized proteomic to identify host proteins and explore the complex host cellular responses to viral infection [8,9]. Additionally, investigation of the changed genes or proteins using the proteomics upon virus infection host cells has become an effective instrument for several viral pathogens, including duck hepatitis A virus type 1 [10], *Bombyx mori* nucleopolyhedrovirus [11], caprine parainfluenza virus type 3 [12], influenza virus [13], and Dengue virus [14]. Mass spectrometry (MS)/MS-based method, isotopomer label, and tandem mass tag (TMT) have been widely used for detecting the interaction between hosts’ protein and virus infection [15]. However, proteomic analysis of PSV-infected cells has never been reported and host proteins involved in PSV replication are largely unknown.

In this study, to monitor the cellular proteins differentially expressed after PSV infection and identify host proteins associated with virus replication, we profiled host proteome changes in PSV-infected PK-15 cells using TMT-coupled MS analysis. As a result, a total of 181 proteins were differentially expressed after PSV infection. Quantitative PCR (RT-qPCR) and parallel reaction monitoring (PRM) were applied to verify proteomic analysis. We found that syndecan-1 (SDC1) is involved in PSV replication. Our studies may provide an opportunity to reveal the relationships of pathogenesis and the interactions between PSV and PK-15 cells.

## 2. Results

### 2.1. PSV Infection in PK-15 Cells

To determine the better time for PSV infection in PK-15 cells, cytopathogenic effects (CPEs) and viral titers were monitored at different time points after virus infection. CPEs became visible at 6 hours post infection (hpi). At 8 hpi, an obvious CPE was observed and over 50% of cells fell off at 10 hpi (Figure 1A). The one-step growth curve showed that the viral titer reached 4.59 × log_10_ 50% tissue culture infective doses (TCID50)/100 μL at 6 hpi, peaked at 5.289 × log_10_TCID50/100 μL at 8 hpi, and then declined (Figure 1B). Additional data obtained from Western blot analysis revealed that the virus protein (VP)1 expression level was high at 8 hpi (Figure 1C). Therefore, to better understand protein change of PSV-infected PK-15 cells, we chose 8 hpi as the time points for further proteomic analysis.

### 2.2. Identification of Differentially Expressed Proteins (DEPs)

Protein extracts from mock-infected and PSV-infected samples were subjected to TMT-coupled LC–MS/MS analysis and a total of 5690 proteins were identified (Appendix A). Additionally, we obtained 122,204 spectra and 45,938 identified peptides (38,917 unique peptides) (Appendix A). Based on the following criteria, fold-change ratios ≥ 1.2 or ≤0.833 and a *p* < 0.05, we found 181 DEPs including 59 up-regulated and 122 down-regulated DEPs (Figure 2A, Appendix A). Among them, the top 20 DEPs are shown in Table 1. Additionally, a heatmap based on the UniProt database was constructed with red representing up-regulated DEPs and green indicating down-regulated DEPs (Figure 2B). The hierarchical clustering classified the DEPs into two categories with opposite directional variation, suggesting the screening rationality of DEPs.

### 2.3. Functional Classification and Enrichment Analysis of the Identified DEPs

To further study the impact of DEPs in cell physiological process, all 181 DEPs underwent gene ontology (GO) enrichment analysis (Appendix A). Based on *p* value in the highest level, the top 20 enriched GO terms are shown in Figure 3A. In the category of biological process (BP), the DEPs were associated with various biological processes, including negative regulation of organelle assembly (5 proteins), regulation of organelle assembly (9 proteins), and regulation of RNA splicing (9 proteins). In the category of molecular function (MF), three dominant enriched functions including nucleic acid binding (58 proteins), RNA binding (36 proteins), and transcription regulator activity (17 proteins) were identified. In cellular component (CC) classification, nucleus (76 proteins), myosin complex (6 proteins), and endosomal sorting complex required for transport (ESCRT III) complex (4 proteins) were the major components of the identified proteins.

We then performed Kyoto Encyclopedia of Genes and Genomes (KEGG) pathway analysis to further investigate the pathways of the identified DEPs. We found that DEPs were mainly involved in 6 signaling pathways, including extracellular matrix (ECM)-receptor interaction (P14287, Q8HZJ6, K7GNF5), fluid shear stress and atherosclerosis (A0A287AD21, P61958, A7WLH8, Q8HZJ6, K7GNF5), RNA transport (A0A286ZWZ7, A0A287AD21, P61958, A7WLH8, A0A287BF68), thyroid hormone synthesis (P50390 and F1RZR0), retinoic acid–inducible gene I (RIG-I)-like receptor signaling pathway (A0A287AB58, F1S8C6), and cell adhesion molecules signaling pathway (Q8HZJ6 and K7GNF5) (Figure 3B, Appendix A).

After analyzing the subcellular localizations of the DEPs, the DEPs were mainly distributed in the nucleus (70.33%) and cytoplasm (16.75%), followed by extracellular space (4.78%), mitochondria (2.87%), and plasma membrane (2.87%). Cytoskeleton (1.91%) and a smaller portion were localized in the chloroplast (0.48%) (Figure 4, Appendix A). More detailed information for subcellular localizations of the DEPs can be obtained in Appendix A.

Based on the DEPs, we constructed a protein–protein interaction network and found that 39 DEPs were mapped to five protein–protein interaction networks (Figure 5, Appendix A). In all these protein–protein interaction networks, the top three DEPs with the highest node degree were A0A287AIS0 (13), A0A287A2U0 (11), and I3LQS0 (11), suggesting that these proteins may be the important candidate proteins for research on PSV infection in PK-15 cells.

### 2.4. Validation of DEPs by PRM and qRT-PCR

To confirm the DEPs via LC- mass spectrometry (MS)/MS data, we first performed parallel reaction monitoring (PRM) analysis for four selected DEPs (two up-regulated and two down-regulated proteins). As shown in Figure 6, the relative abundance of DEPs is presented as a box plot, with identical expression level to the proteomic data. Moreover, no noticeable differences were found of the virus/control ratio between PRM and tandem mass tag (TMT)/MS. All these results suggested that our TMT/MS analysis was reliable.

Additionally, we selected 5 representative proteins osteopontin (SPP1), induced protein with tetratricopeptide repeats 5 (IFIT5), ISG15 ubiquitin-like modifier (ISG15), vinculin (VCL), and syndecan-1 (SDC1) to confirm the proteome analysis and study the kinetics by qRT-PCR. These proteins were chosen in the light of interest and for their different ratios. As shown in Figure 7, increased transcript levels of SPP1, IFIT5, ISG15, and VCL and decreased transcript level of SDC1 were found after PSV infection of 4 h and 8 h, which were consistent with the results of LC-MS/MS data.

### 2.5. SDC1 Promotes PSV Replication 

SDC1 is a cell surface proteoglycan and can mediate host defense mechanisms, angiogenesis, and virus attachment and entry [16]. During porcine hemagglutinating encephalomyelitis virus infection, SDC1 acts as a target gene of miR-10a-5p and short interfering RNA (siRNA)-mediated knockdown of SDC1 reduces viral replication [17]. To examine the effect of the differentially expressed proteins on PSV infection, we transfected HA-tagged SDC1 into PK-15 cells. At 24 h post-transfection, cells were infected with PSV and VP1 protein expression level was analyzed using western blot. We found that VP1 protein level was significantly increased by SDC1 over-expression in PK-15 cells compared with the transfected pcDNA-HA group (Figure 8A). To further confirm our results, siRNAs against SDC1 were transfected into PK-15 cells for 24 h, then cells were harvested, and the knockdown efficiency was tested by qRT-PCR. As shown in Figure 8B, siRNAs could decrease intracellular mRNA levels of targeted SDC1. PK-15 cells were then transfected with siSDC1 or siNC, followed by a virus entry assay. As expected, reduction of SDC1 expression by RNA interference could significantly decrease VP1 protein synthesis compared to siNC-treated cells and in a dosage-dependent form (Figure 8C). As a complementary approach, viral titer assays indicated that the virus titer of PK-15 cells transfected with HA-SDC1 (6.564 × log_10_TCID50/100 μL) was significantly increased compared to that of pcDNA-HA transfected cells (5.184 × log_10_TCID50/100 μL, Figure 8D). Knockdown of SDC1 significantly increased PSV titer compared to that in siNC-treated cells (*p* < 0.01, Figure 8D). All together, these findings indicate that SDC1 could promote PSV replication.

## 3. Discussion

PSV has been regarded as a causative agent of neurological, diarrheal, and fertility disorders in pig. Although the PSV receptor and entry pathway have been reported [5,18], the information of target proteins related to virus infection is largely unknown. Comparative proteomic analysis involving TMT has been widely used to explore host responses of various viral and fungal infections [9,19,20]. Proteomic methods in virology play important roles in revealing virus replication, antiviral responses, and virus–host protein interactions at the protein level [19,21,22]. In this study, we identified that the host proteins, IFIT5, ISG15, SPP1, and VCL, were up-regulated and SDC1 was down-regulated during PSV infection using RT-qPCR.

IFIT proteins contain repetitive sequences of the regulatory tetratricopeptide repeat domain, which mediates protein–protein interactions, including translation initiation, viral replication, and double-stranded RNA signaling [23,24]. IFIT1 can combine with human papillomavirus protein E1, preventing viral replication in the nucleus [25]. Knockout of chicken IFIT5 fibroblasts using CRISPR/Cas9 increases Newcastle disease virus and vesiculitis virus replication [26]. ISG15 plays important roles in antiviral response and is involved in the regulation of various viruses, including influenza virus, vaccinia virus, and human respiratory syncytial virus [27,28,29]. In the early stages of pseudorabies virus (PRV) infection, pig ISG15 mRNA and protein expression levels are increased, and play important roles in antiviral effect and regulating PRV replication [30]. Additionally, ISG15 levels are significantly elevated in patients with chronic hepatitis C virus and hepatitis E virus infection [31,32]. In the current study, using proteomics analysis we found significant increases of IFIT5 and ISG15 in PSV-exposed cells, indicating that IFIT5 and ISG15 may be pivotal regulatory factors in PSV infection.

As a fibrogenic cytokine, SPP1 is significantly up-regulated in the process of tissue damage and repair [33]. In the primate models of HIV infection, viral infection causes a significant increase of SPP1 expression in the brain and cerebrospinal fluid [34]. In addition, activation of SPP1 promotes the replication of hepatitis C virus [35] and the neural invasion of West Nile virus [36]. VCL, a 116- kDa cytoskeletal protein, is involved in the connection between integrin adhesion molecules and actin cytoskeleton [37]. VCL levels are increased in lymphoid cells and peripheral blood mononuclear cells of human immunodeficiency virus type 1 (HIV-1)-infected patients [38]. Overexpression of VCL reduces the susceptibility of human cells to HIV-1 and Moloney mouse leukemia virus infection [39]. In this study, we found that SPP1 and VCL were up-regulated in PSV-infected cells, suggesting that SPP1 and VCL appear to play an important role in the process of PSV infection.

Syndecan is a family of transmembrane heparan sulfate (HS) during the maturation of B cells [40]. SDC1 is considered to be an important marker for the diagnosis and prognosis of AIDS-related lymphoma [41]. SDC1 represents the dominant HS expressed on the surface of multiple myeloma cells and regulates cell growth and survival [42]. The function of SDC1 is regulated by intracellular transport, and Rab5 can bind to SDC1 in the cytoplasmic domain and mediate its internalization [43]. SDC1 is also the major receptor for herpes simplex virus type-1 attaching to cells [44]. Changes in SDC1 expression levels could be observed during infection with *Pneumocystis* and *Neisseria gonorrhoeae* [45,46]. Additionally, Anastasiadou et al. [47] found that Epstein-Barr virus infection causes down-regulation of SDC1. RNA interference silencing SDC1 can reduce the entry of herpes simplex virus type 1 into HeLa cells, inhibit the formation of viral plaques, and promote cell survival [44]. Similar to these studies, we also found that SDC1 expression can be significantly down-regulated during PSV infection. Overexpression of SDC1 can increase the expression of PSV VP1 protein and viral titer, while knockout of SDC1 has the opposite results.

SDC1 is a cell surface HS proteoglycan that is mainly expressed in epithelia [48]. SDC1 ectodomain includes polysulfated HS chains that facilitate interactions with many proteins, including viruses, growth factors, and chemokines [49]. A previous study showed that classical swine fever virus (CSFV) infects swine kidney cells using cellular membrane-associated HS to facilitate viral entry [50], and viral glycoprotein E^rns^ interacts with HS, causing CSFV infection in a HS-dependent mechanism [51]. At early stages of hepatitis C virus infection, SDC1 and virions colocalize at the plasma membrane and are internalized in endosomes of hepatocyte, and knocking down SDC1 inhibits HCV infection [52]. However, as a cell surface proteoglycan being extensively glycanated with HS, whether SDC1 regulates PSV replication caused by HS and how SDC1 affects the PSV life cycle need to be further investigated.

In conclusion, using proteomic analysis we found that the DEPs SPP1, IFIT5, ISG15, VCL, and SDC1 are significantly changed in PSV-infected PK-15 cells, suggesting that these proteins may be closely related to viral infection. Findings of these studies may provide a new reference for the prevention and control of PSV.

## 4. Materials and Methods

### 4.1. Cell Culture and Virus

PK-15 cells were cultured in Dulbecco’s minimal essential medium (DMEM) supplemented with 10% FBS (Gibco, Grand Island, NY, USA) at 37 °C. The PSV strain csh was isolated and preserved by our laboratory [4]. Virus stock was propagated in PK-15 cells and the virus titer was determined by the Reed–Muench method.

### 4.2. Virus Titration

PSV was added to PK-15 cells at a multiplicity of infection (MOI) = 2, and after 1.5 h of adsorption, culture medium was removed. Then the cells were washed twice with PBS, and fresh medium was added. For growth kinetics, at the indicated time post-infection, cytopathic effect was recorded and virus titers were calculated using the Reed–Muench method and recorded as TCID50/100 μL. For virus titration detection, PK-15 cells were transfected with SDC1 overexpression, SDC1 knockdown (30 pmol), or negative controls (pcDNA-NC or siNC). After transfection for 24 h, cells were infected with PSV (MOI = 2). At the indicated time, cells were harvested and cell lysates were collected for TCID50 analysis.

### 4.3. Protein Extraction

For quantitative proteomic analysis, PK-15 cells cultured in 150-mm cell culture dishes were infected with PSV (MOI = 2) or mock treated. At 8 h post-infection (hpi), cells were washed with prechilled PBS, collected with cell scrapers, resuspended in 500 μL of lysis buffer containing 1 mM protease inhibitor cocktail, incubated for 10 min on ice with gentle agitation, and centrifuged at 4000× *g* for 5 min at 4 °C.

### 4.4. Tandem Mass Tag (TMT) Labeling

For TMT labeling, the high-abundance proteins in the cell samples were removed using ProteoSeek™ Albumin/IgG removal kit (Thermo Fisher Scientific, Rockford, IL, USA) according to the manufacturer’s instructions. The cell samples were then extracted using SDT lysis buffer (4% SDS, 100 mM Tris/HCl pH = 7.6, 0.1 M DTT) [53]. Protein concentration in extracts was measured with the Bradford assay (Bio-Rad, Hercules, CA, USA), followed by filter-aided sample preparation digestion for peptide extraction with trypsin, as previously reported [53]. Peptides were then quantified spectrophotometrically at OD_280_, and 100 μg of each peptide sample was labeled with unique isobaric TMT reagents using TMT Mass Tagging kit (Thermo Fisher Scientific). Peptides of the mock-treated group for three biological repeats were labeled as TMT-126, TMT-127, and TMT-128, respectively, and three biological repeats of the virus group were labeled as TMT-129, TMT-130, and TMT-131, respectively.

### 4.5. High pH Reversed-Phase Fractionation

The labeled peptides of each group were equivalently mixed and then fractionated using high pH reversed-phase peptide fractionation kit (Thermo Fisher Scientific), as previously reported [54]. Briefly, the reversed-phase fractionation spin columns were first equilibrated with acetonitrile and 0.1% trifluoroacetic acid, followed by loading the labeled peptides into the columns. After adding ultrapure water into the columns, low-speed centrifugation was carried out for desalination. For desalination treatment, ultrapure water was added into the columns, followed by low-speed centrifugation. Then, gradient acetonitrile was used for fractions’ elution. The labeled peptides were finally collected, dried with a vacuum concentrator, and dissolved with 12 μL 0.1% formic acid. The concentration of each fractionation sample was quantified at an absorbance of OD_280_.

### 4.6. Liquid Chromatography–Mass Spectrometry (LC–MS/MS) Analysis

Reverse-phase separation was performed on a HPLC system (Easy nLC 1200, Thermo Scientific). The peptide mixture was loaded onto a reverse phase trap column (Thermo Scientific Acclaim PepMap100, 100 μm × 2 cm, nanoViper C18) connected to the C18-reversed phase analytical column (Thermo scientific EASY column, 10 cm, ID75 μm, 3 μm, C18-A2) in buffer A (0.1% formic acid) and separated with a linear gradient of buffer B (84% acetonitrile and 0.1% formic acid) at a flow rate of 300 nL/min by Intelli Flow technology.

The separated samples were analyzed using Q-Exactive mass spectrometry and Easy nLC (Proxeon Biosystems, now Thermo Fisher Scientific) for 1 h with positive ion mode as the mass spectrometer. Full MS scans were acquired in the mass range of 300–1800 m/z with a mass resolution of 70,000 at 200 m/z, an automatic gain control target value of 10^6^, a maximum inject time of 50 ms, and dynamic exclusion duration of 60 s. Survey scans were acquired at a resolution of 70,000 at 200 m/z and resolution for higher energy collisional dissociation (HCD) spectra was set to 17,500 at 200 m/z, and the isolation width was 2 m/z. Normalized collision energy was 30 eV and the underfill ratio, which specifies the minimum percentage of the target value likely to be reached at maximum fill time, was defined as 0.1%.

### 4.7. Protein Identification and Quantitation

All MS raw data files were analyzed by Proteome Discoverer software 1.4 (Thermo Fisher Scientific) using the Mascot 2.2 search engine against a database of *Mustela putorius* furo protein sequences (NCBInr, released on 23 March 2017, containing 38,992 sequences). For protein identification, a fragment mass tolerance of 0.1 Da and with permission of two missed cleavages in the digests were allowed. Carbamidomethylation (C), TMT 6/10 plex (N-terminal), and TMT 6/10 plex (lysine, K) were set as the fixed modifications. The oxidation (methionine, M) and TMT 6/10 plex (tyrosine, Y) were set as the variable modifications, with peptide mass tolerances at ± 20 ppm for all MS1 spectra acquired and the value of false discovery rate was ≤ 0.01 [55]. A protein ratio was expressed as a median value of only unique peptides of the protein. All peptide ratios were normalized by the median protein ratio, and the median protein ratio should be “1” after the normalization.

### 4.8. Bioinformatics Analyses

Hierarchical clustering analysis of the differentially expressed proteins was performed based on Cluster 3.0 (http://bonsai.hgc.jp/~mdehoon/software/cluster/software.htm) and Java Treeview software (http://jtreeview.sourceforge.net) [56]. Euclidean distance algorithm for similarity measures and a clustering algorithm for averaged linkage (the centroids of the observations are necessary for clustering) were chosen during the process of hierarchical clustering. Gene ontology (GO) enrichment on three ontologies (biological process (BP), molecular function (MF), and cellular component (CC)) was applied based on Blast2GO (http://www.blast2go.de) [57]. Kyoto Encyclopedia of Genes and Genomes (KEGG) pathway annotation was extracted from the KEGG Automatic Annotation Server software [58]. The enrichment analyses for GO and KEGG annotations were performed using the Fisher’s exact test. Based on STRING (http://string-db.org/) database [59], the protein–protein interaction networks of the altered proteins were constructed and visualized using CytoScape software (version 3.2.1) [59]. The workflow of the LC-MS/MS and bioinformatics analyses is listed in Appendix A.

### 4.9. PRM Analysis

Based on the proteome results, four proteins (two up-regulated and two down-regulated) were selected for PRM validation. After the peptide samples were prepared, they were analyzed using Q-Exactive HF Plus (Thermo Fisher Scientific). A full mass spectrum was detected using the Orbitrap at a resolution of 60,000 (scan range: 300–1800 m/z), followed by 200 MS/MS scans using the Orbitrap at a resolution of 30,000 (200 m/z, automatic gain control target of 3 × 10^6^, maximum injection time of 120 ms, normalized collision energy of 27) in a data-independent procedure. The raw data of PRM were analyzed using Skyline 3.5.0 software.

### 4.10. The qRT-PCR

Total cellular RNA was extracted using TRIzol (Thermo Fisher Scientific) according to the manufacturer’s instructions. The cDNA was reverse-transcribed from 1 μg of total RNA using oligo(dT) primers (Vazyme Biotech Co., Nanjing, China). The qRT-PCR was carried out using SYBR Green master mix (Vazyme Biotech Co.) and specific primer (Appendix A). Relative transcript levels were calculated using the ΔΔC_t_ method. The relative expression values of the targeted gene were normalized to the expression value of GAPDH.

### 4.11. Plasmid and RNA Interference Transfection

The porcine SDC1 were amplified from the cDNA obtained from PK-15 cells. Plasmid expressing HA-tagged SDC1 was constructed by our laboratory and sequenced by Sangon (Shanghai, China). Small interfering RNAs (siRNAs) against *Sus scrofa* SDC1 (siSDC1, 5′-GGAAACCGUGGCCACAAAUTT-3′) and negative control (siNC, 5′-UUCUCCGAACGUGUCACGUTT-3′) were synthesized by GenePharma (Shanghai, China). PK-15 cells were first seeded on 24-well plates, plasmid or siRNA transfection was then performed with lipofectamine 6000 (Beyotime Biotechnology, Shanghai, China) according to the manufacturer’s instructions. The knockdown efficiencies of siSDC1 were quantified by RT-qPCR. At 28 h post-transfection, cells were infected with PSV (MOI = 2) for 8 h, and virus replication was detected by Western blot.

### 4.12. Western Blotting

Extracted protein mixtures were loaded into SDS gel and electrotransferred onto PVDF membranes. The membrane was blocked with 5% nonfat milk and incubated with mouse anti-PSV VP1 antibody (1:1000) or anti-HA antibody (1:5000) at 4 °C overnight. After washing for three times, the membrane was incubated with secondary antibodies conjugated to HRP (1:10000) for 1 h. The α-tubulin was used as a loading control. Finally, bands were developed with ECL prime Western blot detection reagent (GE Healthcare) and then quantified with Image Pro-Plus software.

### 4.13. Statistics

Statistical analysis was performed by SPSS version 22.0 (SPSS Inc., Chicago, IL, USA). Data are expressed as mean ± SD for at least two independent experiments. Statistical significance was tested using Student’s *t*-test to compare two groups and analysis of variance with Tukey’s post hoc test to compare more groups. The *p* < 0.05 was regarded as significant difference.

## Figures and Tables

**Figure 1 ijms-21-04386-f001:**
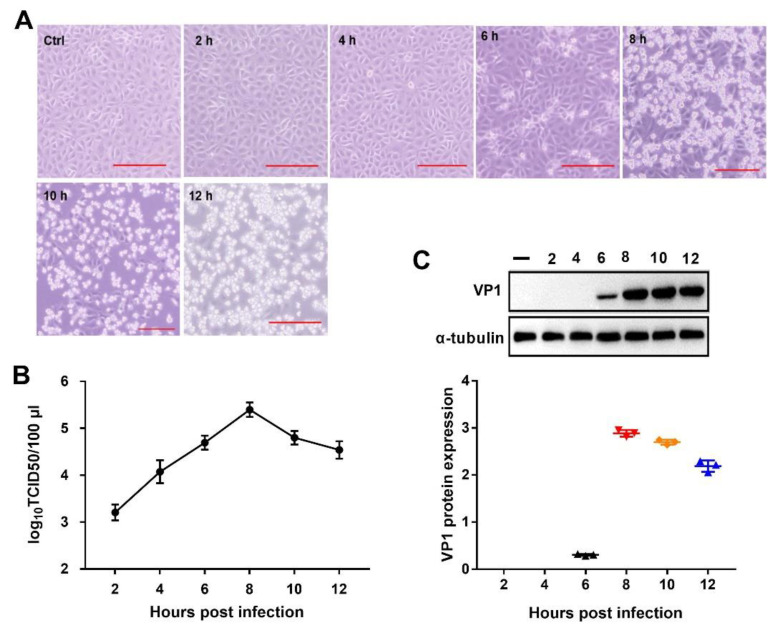
Sapelovirus A (PSV) infection in pig kidney (PK)-15 cells. (**A**) PK-15 cells were mock-treated or treated with PSV at a multiplicity of infection (MOI) of 2, and the cytopathogenic effects at indicated time were observed. Bars, 50 μm. (**B**) One-step growth curve of PSV in PK-15 cells. (**C**) Western blotting assay of PSV virus protein (VP)1 expression in PK-15 cells at the indicated time points (2–12 h). The α-tubulin was used as a sample loading control. Data are expressed as mean ± SD for two independently experiments.

**Figure 2 ijms-21-04386-f002:**
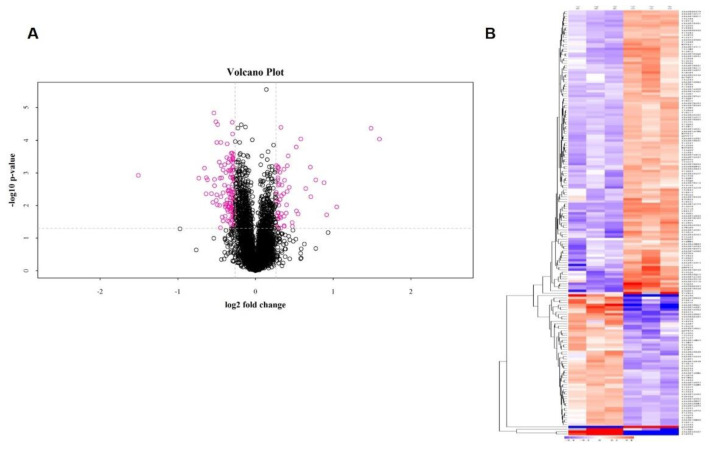
Volcano plot and the hierarchical clustering involved in the differentially expressed proteins (DEPs). (**A**) Volcano plot showed the levels of DEPs detected in mock-treated or PSV-infected groups. X axis, mean log_2_(ratio of fold change); y axis, log_10_(*p*-value); red balls, DEPs between the two groups (*p* < 0.05 and ratio > 2 or <0.5). (**B**) Hierarchical cluster of 181 DEPs with 1.2-fold up-regulation and 0.85-fold down-regulation. The proteins were classified into two categories with opposite directional variation, suggesting the screening rationality of DEPs. Different color indicated different fold change. Red: Highly expressed, blue: Lowly expressed, and white: No change.

**Figure 3 ijms-21-04386-f003:**
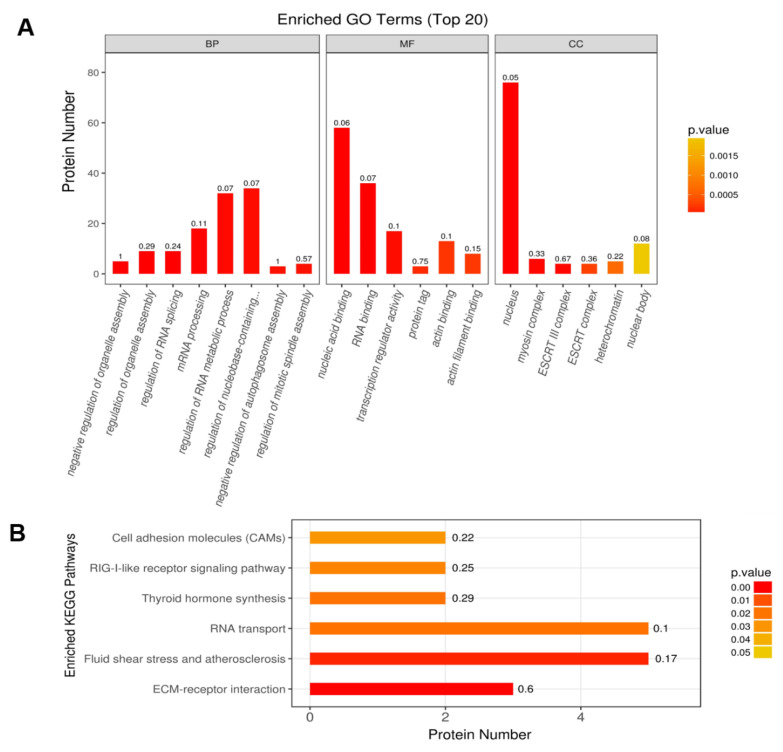
Top 20 enriched, differentially expressed proteins processed of gene ontology (GO) terms and enriched Kyoto Encyclopedia of Genes and Genomes (KEGG) pathways in PSV-infected PK-15 cells. (**A**) GO annotation of the DEPs. BP: Biological process; MF: Molecular function; CC: Cellular component. (**B**) Enriched KEGG pathways. The number on the top of each column represents enrichment factor. The color of the bar chart indicates the *p* value based on Fisher’s exact test.

**Figure 4 ijms-21-04386-f004:**
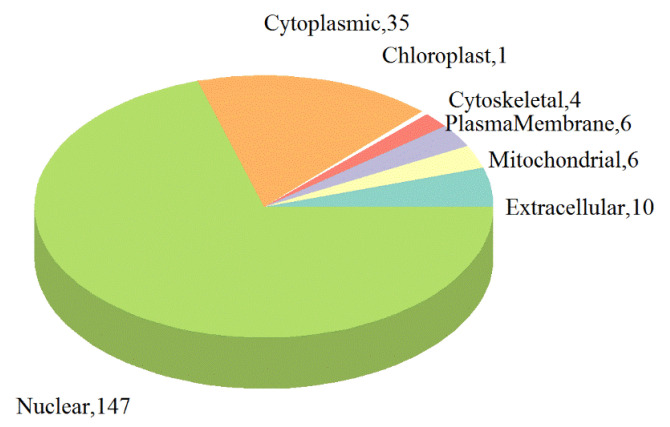
Subcellular locations of the DEPs in PK-15 cells infected with PSV.

**Figure 5 ijms-21-04386-f005:**
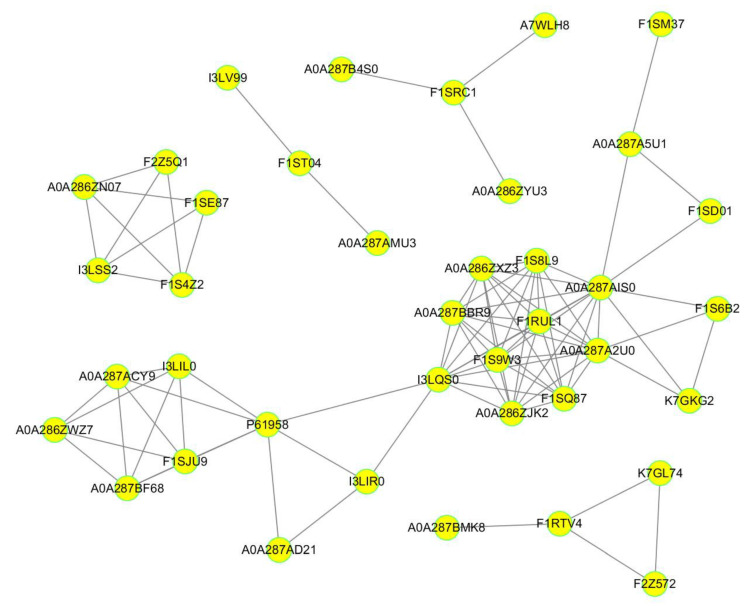
Protein-protein interaction networks were constructed based on the DEPs. The yellow nodes indicate the DEPs, and the lines are protein-protein interactions.

**Figure 6 ijms-21-04386-f006:**
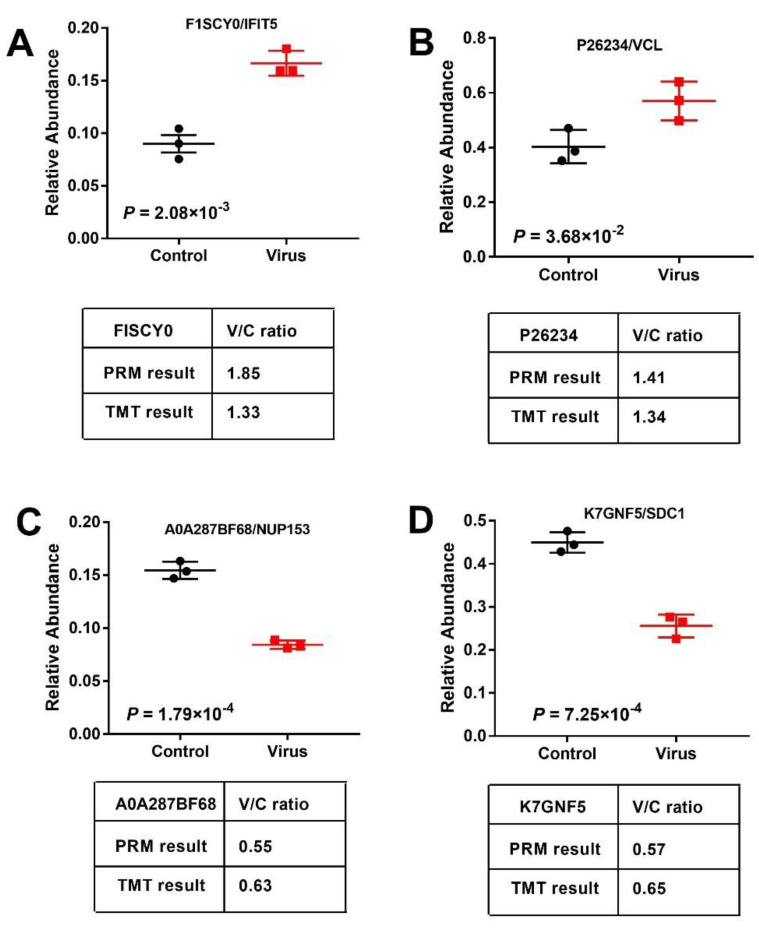
DEPs including induced protein with tetratricopeptide repeats 5 (**A**, IFIT5), vinculin (**B**, VCL), nucleoporin (**C**, NUP153), and syndecan-1 (**D**, SDC1) were verified using PRM and the virus/control (V/C) ratio was compared with the TMT result. Data are expressed as mean ± SD for three replicates.

**Figure 7 ijms-21-04386-f007:**
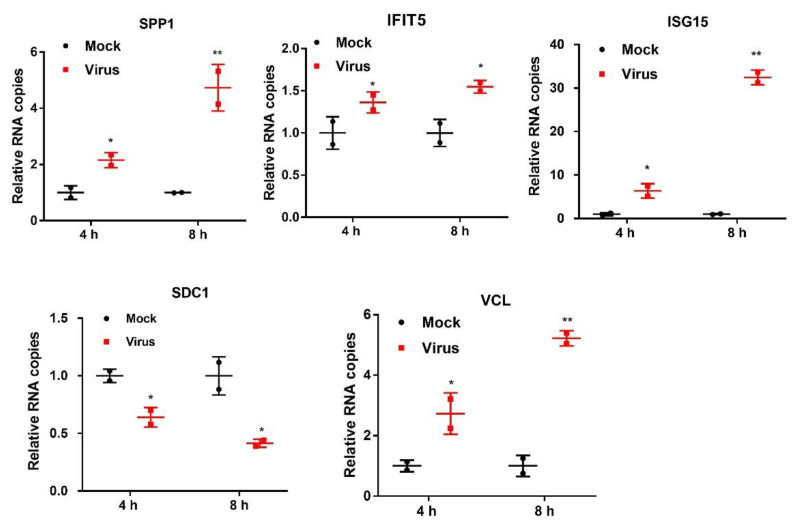
The expression of the indicated genes and kinetics in PK-15 cells infected with PSV for 4 h and 8 h was measured by RT-qPCR. Data are shown as mean ± SD for two experiments; * *p* < 0.05, ** *p* < 0.01.

**Figure 8 ijms-21-04386-f008:**
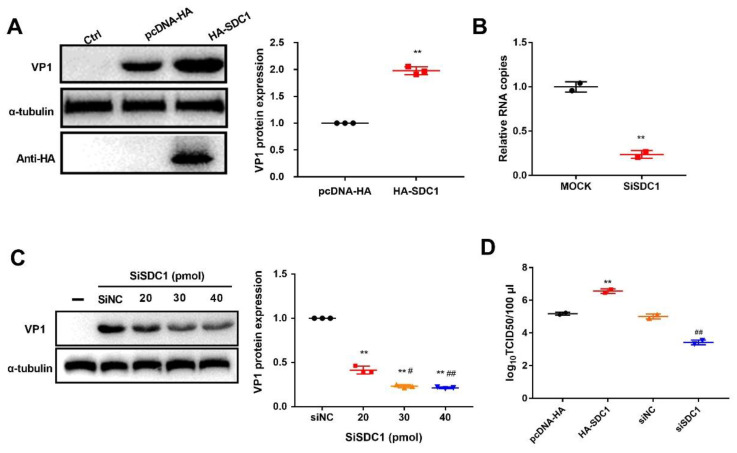
The effect of syndecan-1 (SDC1) on PSV replication. (**A**) After 24 h transfection with pcDNA-HA empty or HA-SDC1, PK-15 cells were infected with PSV (MOI = 2), cells were lysed, and virus VP1 proteins were analyzed by Western blot. (**B**) The effect of siRNA against SDC1 was determined by RT-qPCR. (**C**) PK-15 cells were untreated or transfected with siNC, 20 pmol, 30 pmol, or 40 pmol siSDC1. At indicated time after transfection, cells were infected with PSV and processed for Western blot. (**D**) Virus titers were determined by 50% tissue culture infective doses (TCID50) of samples prepared at 8 h post-infection from PK-15 cells infected with PSV after transfection with pcDNA-HA empty, HA-SDC1, siNC, or 30 pmol siSDC1. Two independent experiments were performed. The ** *p* < 0.01 compared with siNC or pcDNA-HA group. The ^#^
*p* < 0.05, ^##^
*p* < 0.01 compared with 20 pmol group.

**Table 1 ijms-21-04386-t001:** The significantly up-regulated and down-regulated differentially expressed proteins (top 20) in PK-15 cells infected with Sapelovirus A.

Accession	Description	Coverage	Unique Peptides	Fold-Change Ratios	*p*-Value
**Up-regulated**					
A0A286ZSR7	Olfactory receptor	2.56	1	3.02	9.2167 × 10^−5^
F1RTV4	Amidophosphoribosyltransferase	1.9	1	2.80	4.2788 × 10^−5^
I3LEQ6	Ral GTPase activating protein catalytic alpha subunit 2	2.13	1	2.06	0.0111
F1RL90	PPARG coactivator 1 beta	1.56	1	1.88	0.0194
A0A287BN67	Nucleosome assembly protein 1 like 1	21.93	1	1.84	0.0020
A0A287AZH1	Nucleosome assembly protein 1 like 4	27.73	9	1.71	0.0017
A0A287ATX4	Wiskott-Aldrich syndrome like	1.58	1	1.64	0.0054
A0A286ZN07	Charged multivesicular body protein 4A	10.36	2	1.63	0.0007
F1RRI4	Sjogren syndrome/scleroderma autoantigen 1	7.54	2	1.56	0.0030
F1SU38	Urokinase-type plasminogen activator	16.29	6	1.50	9.07 × 10^−5^
S5A7T7	SP110 nuclear body protein variant 2	8.67	1	1.46	0.0181
F2Z572	Phosphoribosyl pyrophosphate synthetase 1	18.55	2	1.44	0.0146
A0A0B8RSH5	Nuclear mitotic apparatus protein 1	45.07	2	1.44	0.0002
F1RYV0	Cyclin dependent kinase like 2	1.23	1	1.39	0.0438
A0A287BRG2	Required for excision 1-B domain containing	4.15	1	1.39	0.0333
O77637	Transcription factor NFAT (Fragment)	15.83	2	1.35	0.0028
P26234	Vinculin	55.15	59	1.34	0.0028
F1SCY0	Interferon induced protein with tetratricopeptide repeats 5	6.02	3	1.33	0.0006
Q4VK70	Glycogen [starch] synthase (Fragment)	3.91	2	1.33	0.0037
A0A287AB58	Ubiquitin-like modifier	19.16	3	1.32	0.0095
**Down-regulated**					
Q56PB8	RPGR (Fragment)	1.38	1	0.35	0.0012
I3LQ36	Uncharacterized protein	7.38	1	0.60	0.0015
A0A287BF68	Nucleoporin 153	14.79	18	0.63	0.0007
A0A0B8RSR5	Interleukin enhancer binding factor 3	36.61	1	0.64	0.0044
K7GNF5	Syndecan	4.85	1	0.65	0.0014
F1SB63	T-complex protein 1 subunit alpha	54.92	1	0.65	0.0016
A0A286ZQL5	NOVA alternative splicing regulator 2	1.63	1	0.68	0.0043
A0A287BPS0	SET domain containing 2	1.4	3	0.68	0.0080
A0A287ACY9	Uncharacterized protein	4.93	4	0.69	1.4560 × 10^−5^
F1SFU1	PHD finger protein 23	3.31	1	0.69	0.0016
A0A286ZVL0	Heterogeneous nuclear ribonucleoprotein U	45.21	3	0.69	0.0025
I3LIL0	Nucleoporin like 2	4.31	1	0.70	2.6992 × 10^−5^
Q8HZJ6	Syndecan-4	35.64	5	0.70	0.0070
C8ZL57	Activating transcription factor 6 (Fragment)	10.75	1	0.70	0.0001
F1S8L9	Heterogeneous nuclear ribonucleoprotein U	44.65	2	0.70	0.0047
F1SL26	Tumor necrosis factor receptor superfamily member 1A	2.63	1	0.71	0.0096
F1SDY8	Zinc finger CCCH-type containing 14	24.5	14	0.71	3.3193 × 10^−5^
A0A286ZJK2	Heterogeneous nuclear ribonucleoprotein M	55.86	39	0.72	0.0005
I3LLM2	Olfactory receptor	2.31	1	0.73	0.0010
F1SD01	Cleavage stimulation factor subunit 2 tau variant	11.87	3	0.73	0.0014

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
