# Peer review of "Proteome Analysis Reveals Syndecan 1 Regulates Porcine Sapelovirus Replication"

_ijms, 2020, doi:10.3390/ijms21124386_

Round 1
Reviewer 1 Report
The authors present a very interesting proteome analysis of PK-15 target cells infected with porcine sapelovirus. Despite the soundness of the proteomic and bioinformatic analyses, this reviewer in unclear about the potential role of syndecan-1 (SDC1) in the virus' pathogenesis.
SDC1 is a cell surface proteoglycan being extensivele glycanated with heparan sulfate and chondroitin sulfate chains. The authors do not mention anywhere in their manuscript the potential consequences on glycan patterning (of the ECM) on PK-15 target cells, if any!?
The study should be supplemented by FACS analyses relating the functional expression of SDC1 on the target cells.
The kinetics of up- or down-regulation of the proteins post-transfection should be investigated, which is especially important for proteins with different half lifes of mRNA and protein product.
The PK-15 are porcine kidney cells: wouldn't the study have profited from a comparison with porcine lung cells, which are derived from the prime target organ of the virus? The authors should at least comment on that.
Author Response
Dear Editor,
Thank you for your decision letter concerning our manuscript (ID ijms-770805) entitled "Proteome Analysis Reveals Syndecan 1 Regulates Porcine Sapelovirus Replication", and your time regarding for our revision. I also appreciate all the critical comments from you and reviewers. We have carefully considered the comments and revised the manuscript accordingly. With these improvements, we hope that the current version can meet the Journal’s standards for publication. The following is a point-by-point response to all those comments and a list of changes we have made to the manuscript.
Sincerely
Xiuguo Hua
Point-by-point responses to the comments of the Editor and reviewers, and a list of changes are:
Reviews 1:
1.The authors present a very interesting proteome analysis of PK-15 target cells infected with porcine sapelovirus. Despite the soundness of the proteomic and bioinformatic analyses, this reviewer in unclear about the potential role of syndecan-1 (SDC1) in the virus' pathogenesis. SDC1 is a cell surface proteoglycan being extensive eleglycanated with heparan sulfate and chondroitin sulfate chains. The authors do not mention anywhere in their manuscript the potential consequences on glycan patterning (of the ECM) on PK-15 target cells, if any!?
Response: We agree with the reviewer. We have revised the manuscript carefully to add the information of heparan sulfate in the fourth paragraph of Discussion section as follows:SDC1 is a cell surface HS proteoglycan which is mainly expressed in epithela [51]. SDC1 ectodomain includes polysulfated HS chains which facilitates interactions with many proteins, including viruses, growth factors, and chemokines [52]. A previous study has shown that classical swine fever virus (CSFV) infects swine kidney cells using cellular membrane-associated HS to facilitate viral entry [53], and viral glycoprotein Erns interacts with HS, causing CSFV infection in an HS-dependent mechanism [54]. At early stages of hepatitis C virus infection, SDC1 and virions colocalize at the plasma membrane and are internalized in endosomes of hepatocyte, and knocking down SDC1 inhibits HCV infection [55]. However, as a cell surface proteoglycan being extensive eleglycanated with HS, whether SDC1 regulates PSV replication caused by HS and how SDC1 affects the PSV life cycle need to be further investigated.
- The study should be supplemented by FACS analyses relating the functional expression of SDC1 on the target cells.
Response: We would like to thank the respected reviewer for his useful comments, which is helpful for us to clarify the influence of SDC1 on PK-15 cells. The expression of SDC1 on the target cells (PK-15 cells) after PSV infection has been detected using PRM (Fig.6) and TCID50 (Fig.7). As there is no anti-porcine SDC1 antibody suitable for FACS, it is difficult to conduct the experiments of FACS analyses relating the functional expression of SDC1. Thanks for your kind advice again, and probably we may address them later in our next related paper.
- The kinetics of up- or down-regulation of the proteins post-transfection should be investigated, which is especially important for proteins with different half lifes of mRNA and protein product.
Response: That's very good question. In this study, after PSV transfection, we found 59 up-regulated and 122 down-regulated differentially expressed protein (DEPs). We selected 5 DEPs (including 4 up-regulated and 1 down-regulated) and detected these DEPs expression using RT-qPCR after virus transfection 4 h and 8 h (Figure 7) to investigate the kinetics of these proteins. This information has been added in the revised results and figure 7 legends. However, as limited scientific research funds and time, we only selected the proteins we're interested in, and more detailed kinetics of these protein may be investigated in the future.
4.The PK-15 are porcine kidney cells: wouldn't the study have profited from a comparison with porcine lung cells, which are derived from the prime target organ of the virus? The authors should at least comment on that.
Response: Thanks for the good advice. Diseases caused by PSV are characterized mainly by acute diarrhea, polioencephalomyelitis, reproductive disorders and even fatality in pigs [1-3]. The virus could be amplified and secreted in pig kidney cells PK-15, IBRS-2 and LLC-PK [3, 4] and human hepatocarcinoma cells PLC/PRF/5 and HepG2/C3a [5]. Researches of PSV are mainly on PK-15 cells and porcine intestinal epithelial cells and studies on whether PSV can be replication on porcine lung cells are unclear. Your suggestions provide us a good scientific thought.
- Arruda, P., et al., Detection of a novel sapelovirus in central nervous tissue of pigs with polioencephalomyelitis in the USA. Transboundary and emerging diseases, 2017. 64(2): p. 311-315.
- Chen, J., et al., Complete genome sequence of a novel porcine Sapelovirus strain YC2011 isolated from piglets with diarrhea. Journal of virology, 2012. 86(19): p. 10898-10898.
- Lan, D., et al., Isolation and characterization of the first Chinese porcine sapelovirus strain. Archives of Virology, 2011. 156(9): p. 1567-74.
- Kim, D.-S., et al., Porcine sapelovirus uses α2, 3-linked sialic acid on GD1a ganglioside as a receptor. Journal of virology, 2016. 90(8): p. 4067-4077.
- Bai, H., et al., Characterization of porcine sapelovirus isolated from Japanese swine with PLC/PRF/5 cells. Transbound Emerg Dis, 2018. 65(3): p. 727-734.
Reviewer 2 Report
The study by Zhao et al. employed a quantitative proteomics approach to find host proteins whose expression is modulated by the porcine sapelovirus (PSV). From the top five hits, the authors focused their attention on syndecan 1 (SDC1).
Although the data are interesting, conclusions on the role of SDC1 on the viral life cycle are unclear and must be strengthened by an additional experiment.
Major comments
1) In this manuscript, the authors claimed that host protein SDC1 plays a crucial role in PSV replication (lines 22, 75).
a) The authors initially showed that SDC1 expression decreased after PSV infection. They went on to show that while SDC1 overexpression led to increased expression of viral protein 1 (VP1), SDC1 knockdown had the converse effect. Although this shows that SDC1 may play a role in the viral life cycle, it is not clear how these data alone show a central role in replication. To better elucidate the impact of SDC1 on the viral life cycle, the authors must show whether SDC1 knockdown and overexpression have an effect on viral growth and on PSV1’s ability to cause cytopathogenicity (as done on Fig 1A-B).
b) In the results section (sections 2.4 and 2.5), can the authors please justify why they chose to focus on SDC1 out of their top five hits?
c) Lines 94-129 contain a very extensive literature review of SPP1, ISG15, IFIT5 and VCL. While interesting, this can be shortened because the authors chose to focus on SDC1. In fact, I suggest the authors to discuss, in the context of their findings, how host protein SDC1 affects the PSV1 life cycle.
Minor comments
1) Authors should specify the number of biological (i.e. independent experiments) and technical (i.e. number of wells) replicates in all figure captions. Numerical data in bar graphs (Figs. 1, 6 & 7) should be depicted in the form of univariate column scatter graphs (this can be done using GraphPad or R).
Author Response
Dear Editor,
Thank you for your decision letter concerning our manuscript (ID ijms-770805) entitled "Proteome Analysis Reveals Syndecan 1 Regulates Porcine Sapelovirus Replication", and your time regarding for our revision. I also appreciate all the critical comments from you and reviewers. We have carefully considered the comments and revised the manuscript accordingly. With these improvements, we hope that the current version can meet the Journal’s standards for publication. The following is a point-by-point response to all those comments and a list of changes we have made to the manuscript.
Sincerely
Xiuguo Hua
Point-by-point responses to the comments of the Editor and reviewers, and a list of changes are:
Review 2
Major comments
1) In this manuscript, the authors claimed that host protein SDC1 plays a crucial role in PSV replication (lines 22, 75).
- a) The authors initially showed that SDC1 expression decreased after PSV infection. They went on to show that while SDC1 overexpression led to increased expression of viral protein 1 (VP1), SDC1 knockdown had the converse effect. Although this shows that SDC1 may play a role in the viral life cycle, it is not clear how these data alone show a central role in replication. To better elucidate the impact of SDC1 on the viral life cycle, the authors must show whether SDC1 knockdown and overexpression have an effect on viral growth and on PSV1’s ability to cause cytopathogenicity (as done on Fig 1A-B).
Response: We would like to thank the respected reviewer for his useful comments, which is helpful for us to clarify the influence of SDC1 on PK-15 cells. We have detected the role of SDC1 knockdown and overexpression on viral growth using TCID50 in the revised figure 8D.
- b) In the results section (sections 2.4 and 2.5), can the authors please justify why they chose to focus on SDC1 out of their top five hits?
Response: Thanks for your kind advise. We have added the purpose of why we chose to SDC1 for our further study in the results section (sections and 2.5) as follows: SDC1 is a cell surface proteoglycan and can mediate host defense mechanisms, angiogenesis, and virus attachment and entry [16]. During porcine hemagglutinating encephalomyelitis virus infection, SDC1 acts as a target gene of miR-10a-5p and siRNA-mediated knockdown of SDC1 reduces viral replication [17].
- c) Lines 94-129 contain a very extensive literature review of SPP1, ISG15, IFIT5 and VCL. While interesting, this can be shortened because the authors chose to focus on SDC1. In fact, I suggest the authors to discuss, in the context of their findings, how host protein SDC1 affects the PSV1 life cycle.
Response: We would like to thank the respected reviewer for his useful comments, which is helpful for us to clarify the influence of SDC1 on PSV. Thus, in the revised Discussion, we shorten the content of the role of SPP1, ISG15, IFIT5 and VCL on viral replication, and add more information of the influence of SDC1 on PSV life cycle.
Minor comments
1) Authors should specify the number of biological (i.e. independent experiments) and technical (i.e. number of wells) replicates in all figure captions. Numerical data in bar graphs (Figs. 1, 6 & 7) should be depicted in the form of univariate column scatter graphs (this can be done using GraphPad or R).
Response: Regarding the number of biological and technical replicates, the information has been noted in Methods and Figure legends of revised manuscript as follows: Data are expressed as mean ± SD for at least two independently experiments. Furthermore, numerical data in bar graphs of fig1, fig 6, fig 7 and fig 8 have been depicted in the form of univariate column scatter graphs in the revised figures.
Round 2
Reviewer 2 Report
In this revised manuscript, Zhao and colleagues have addressed my concerns appropriately.